# Hybrid HP-BOA: An Optimized Framework for Reliable Storage of Cloud Data Using Hybrid Meta-Heuristic Algorithm

Adnan Tahir [1,2], Fei Chen [1], Bashir Hayat [3], Qaisar Shaheen [2], Zhong Ming [1], Arshad Ahmad [4], Ki-Il Kim [5,*] and Byung Hyun Lim [5]

1   College of Computer Science and Software Engineering, Shenzhen University, Shenzhen 518060, China; adnantahir@szu.edu.cn (A.T.); fchen@szu.edu.cn (F.C.); mingz@szu.edu.cn (Z.M.)
2   Department of Computer Science & IT, The Islamia University of Bahawalpur, Rahim Yar Khan 64200, Pakistan; qaisar.shaheen@iub.edu.pk
3   Department of Computer Science, Institute of Management Sciences, Peshawar 25100, Pakistan; bashir.hayat@imsciences.edu.pk
4   Department of IT & Computer Science, Pak-Austria Fachhochschule: Institute of Applied Sciences and Technology, Haripur 22620, Pakistan; yaarshad@gmail.com
5   Department of Computer Science and Engineering, Chungnam National University, Daejeon 34134, Republic of Korea
*   Correspondence: kikim@cnu.ac.kr

**Abstract:** The prime objective of the cloud data storage process is to make the service, irrespective of being infinitely extensible, a more reliable storage and low-cost model that also encourages different data storage types. Owing to the storage process, it must satisfy the cloud users' prerequisites. Nevertheless, storing massive amounts of data becomes critical as this affects the data quality or integrity. Hence, this poses various challenges for existing methodologies. An efficient, reliable cloud storage model is proposed using a hybrid heuristic approach to overcome the challenges. The prime intention of the proposed system is to store the data effectively in the cloud environment by resolving two constraints, which are general and specific (structural). The cloud data were initially gathered and used to analyze the storage performance. Since the data were extensive, different datasets and storage devices were considered. Every piece of data was specified by its corresponding features, whereas the devices were characterized by the hardware or software components. Subsequently, the objective function was formulated using the network's structural and general constraints. The structural constraints were determined by the interactions between the devices and data instances in the cloud. Then, the general constraints regarding the data allocation rules and device capacity were defined. To mitigate the constraints, the components were optimized using the Hybrid Pelican–Billiards Optimization Algorithm (HP-BOA) to store the cloud data. Finally, the performance was validated, and the results were analyzed and compared against existing approaches. Thus, the proposed model exhibited the desired results for storing cloud data appropriately.

**Keywords:** cloud data storage; cloud computing; resource allocation; virtualization; specific constraints; general constraints; hybrid pelican–billiards optimization





## 1. Introduction

On a cloud network, cloud data are extensively distributed by cloud storage providers. While sharing the data services, the cloud users can share their required information within the group. This thus mitigates the data storage complexity. Furthermore, the users cannot control the storage capacity in a physical manner [1]. Moreover, some flaws can jeopardize the data integrity due to system faults due to the software or hardware and human intervention errors. To combat such problems, cloud storage is a prerequisite when sharing on a cloud network [2]. A user should be blocked from the group or be removed from the group due to misbehavior. Hence, revocation is the standard process

when auditing cloud data for storage. To ensure the security level, data management requires a private key to verify the legitimacy of generating the fileblocks [3]. Through this authentication process, the fileblocks are proven to possess the data. During the user revocation from a group, his/her private key is also removed from the user group. In the general and traditional [4] auditing approaches, the authenticators of revoked users are transformed into the authenticators of the non-revoked cloud user group. For such a scenario, the non-revoked users have to fetch all of the information using the user's fileblocks, which are used to again sign and updatenew authenticators in the cloud network. Because of the high-dimensional representation of cloud data, this process has a high costin terms of computational and overhead communication [5].

Several auditing methods have been developed to further resolve the existing issues, along with user revocation in the storage of cloud data [6]. The revoked user groups are transformed into non-revoked groups, where the private key is again required for authentication. This adds to the computational complexity problem, as it results in more fileblocks [7]. Hence, this has an impact on the cloud environment. In real-world applications, user transformation is critical to achieving better storage performance [8]. Furthermore, the performance is degraded, as this often changes the membership function of the group. Hence, the challenging factor is to design an effective model for real-time data [9]. Depending on the needs of different users, various data files are employed to store the cloud data [10]. To meet the requirements, standard storage products are implemented by service providers to save the data [11]. Thus, it becomes challenging to achieve cost-effective networks and highprovider storage capacity.

In cloud data storage management and virtual machine (VM) assignment, providing high-latency, low-cost, high-quality service and scalability is challenging for researchers. Optimization techniques provide high-quality service. Various technologies have been demonstrated to provide high-quality, reliable cloud data storage, and their challenges and features are illustrated in Table 1. CMPSO [12] provides highly secure and reliable resource allocation over wireless networks and has a low computational cost. However, it does not provide a fine-tuned strategy for accommodating connectivity. Furthermore, the power consumption of the entire system is very high. ANC [13] is easy to implement with increased network performance in terms of robustness and fidelity, and also, the communication throughput is very high. Yet, this strategy is not flexible because of the changing channel qualities in wireless networks, and this decreases the total response time for users when the workload is high. The Tabu meta-heuristic [14] meets the wireless requirements such as heterogeneity, reliability, and lowlatency, and also, it provides high synchronization and updating of data over wireless networks. If an unexpected power outage occurs, the valuable data stored in the data center could be lost and unrecoverable. Hence, there is a high cost to protectthe cloud storage system. OMT [15] provides automatic services when customers require more servicesover the network channel. Hence, it can easily interface with the applications and the data sources. It also has a higher offloading failure probability; therefore, the transmission reliability is decreased. Furthermore, it has less scalability in the search space. The EMSA algorithm [16] is highly elastic, costless, and trustworthy. Moreover, the information is quickly accessible by the users, and it is more reliable. Finally, it has high, virtually limitless storage capacity. Yet, it does not meet the bandwidth requirements and has a low maturity level. Furthermore, it does not have any loop-back connectivity or access control. PKI-based signatures [17] provide greater hardware redundancy and have automatic storage failover. However, the packet loss ratio is very high. In addition, the signal-to-noise ratio is very high during packet transmission. Ant colony optimization (ACO) [18] achieves excellent performance by balancing the network load, providing the increased security and integrity of the information over the network channel. However, it may result in considerable network delays, and it has a high overhead and low service quality in terms of cost, security, and latency. The Fibonacci cryptographic technique [19] can handle the network traffic and has lowcomputational complexity. However, it has a high consumption of network resources,

poornode authentication, a high transmission time, and less caching ability. Hence, to resolve these challenges, a new reliable cloud data storage system was developed with optimization for high-quality service.

**Table 1.** Features and challenges of reliable data storage using optimization.

| Author [Citation] | Methodology | Features | Challenges |
|---|---|---|---|
| Fan et al. [12] | CMPSO | • It provides highly secure and reliable resource allocation over wireless networks.<br>• It has a low computational cost. | • It does not provide a fine-tuned strategy for accommodating connectivity.<br><br>• The power consumption of the entire system is very high. |
| Li et al. [13] | ANC | • It is easy to implement with high network performance in terms of robustness and fidelity.<br>• The communication throughput is very high. | • This strategy is not flexible because of the changing channel qualities in wireless networks.<br>• It decreases the total response time of a user when the workload is high. |
| Assi et al. [14] | Tabu meta-heuristic | • It meets the wireless requirements such as heterogeneity, reliability, and lowlatency.<br><br>• It provides high synchronization and updating of the data over wireless networks. | • If an unexpected power outage occurs, the valuable data stored in the data center could be lost and unrecoverable.<br>• It has a high cost for protectingthe cloud storage system. |
| Zaharie et al. [15] | OMT | • It provides automatic services when the customer requires more services over the network channel.<br>• It can easily interface with the applications and data sources. | • It has a higher offloading failure probability; therefore, the transmission reliability is decreased.<br>• It has less scalability in the search space. |
| Suba et al. [16] | EMSA algorithm | • It is highly elastic, has a lower cost, and is trustworthy.<br>• The information is quickly accessible by the users, and it is more reliable.<br>• It has high, virtually limitless storage capacity. | • It does not meet the bandwidth requirements and has a low maturity level.<br>• It does not have any loop-back connectivity and access control. |
| Shao et al. [17] | PKI-based signature scheme | • It provides greater hardware redundancy.<br><br>• It has the ability of automatic storage failover. | • The packet loss ratio is very high.<br><br>• The signal-to-noise ratio is very high during packet transmission. |
| Akash et al. [18] | ACO | • It achieves better performance by balancing the network load.<br>• It provides high security and integrity of the information over the network channel. | • It may result in large network delays, and it has a high overhead.<br>• The quality of service is low in terms of cost, security, and latency. |
| Sangeetha et al. [19] | Fibonacci cryptographic | • It can handle the network traffic.<br><br>• It provides lowcomputational complexity. | • It has a high consumption of network resources.<br>• It has poornode authentication and a high transmission time.<br>• The caching ability is less. |

Diverse approaches have been deployed to mitigate the cost function and increase the system's reliability [20]. During the storing process of cloud data, some critical issues are met, such as transmission and communication overhead. Scholars have implemented an effective data storage model to alleviate these challenges. Over the past few years, some experts have utilized multi-objective functions to lessen the storage cost and improve the data integrity, even considering the massive amount of data over the cloud network [21,22]. Due to this approach, the best solutions have been obtained to solve the decisive issues to make the storage performance more effective. Moreover, different optimization algorithms such as Particle Swarm Optimization (PSO), the Grey Wolf Optimizer (GWO), etc., have been adopted. In [23], the multi-objective function was derived to maximize the resources

for access and minimize the storage cost, where the performance was measured via the capacity, storage, bandwidth, and so on. In [24], they explored the data migration approach of using virtual machines, and they included the bandwidth of the network, energy optimization, server management, and data centers in the cloud. In [25], they presented a scheduling mechanism to reduce the usage of machines to store the cloud data, in which the efficacy of the model was validated using the bandwidth, storage, and network connection. Hence, effective models have been employed to improve the reliability of cloud data storage. The prime objectives of the paper are summarized as listed below:

- The design and implementation of a novel and reliable cloud data storage scheme to conserve the data with a hybrid meta-heuristic approach to preserve and access the cloud data whenever required.
- Resolving the general and specific constraints of the components to develop a new reliable framework to allocate the cloud data by optimizing the components while allocating the respective components to particular VMs.
- Developing the the novel hybrid meta-heuristic algorithm, named HP-BOA, where the conventional POA is superimposed on the traditional BOA. It is mainly utilized for optimizing the components to provide the desired outcomes.
- Analyzing the model's efficacy with convergence and statistical analysis, where the comparison was performed with other classical optimization algorithms.

The rest part of the paper is organized as follows. The review of the works on the cloud data storage mechanism are elaborated in Section 2. Section 3 elucidates the system model and problem definition of the data storage system. Section 4 specifies the problem and multi-objective functions in the cloud data storage model. Section 5 illustrates the hybrid optimization algorithm for optimizing the components. Section 6 discusses the simulation results and their analyses. The paper's conclusions are given in Section 7.

## 2. Related Work

Fan et al. [12] implemented the "constrained multi-objective particle swarm optimization (CMPSO)" to enhance the reliability of data stored in the cloud. Here, three different costs were considered: data migration, occupation cost in the storage space, and communication cost. To ensure the effectiveness of the storage schemes, the diverse metrics considered were equipment stability, software, and transmission reliability. Finally, the implementation and a comparative analysis were performed. It was evaluated with the three criteria for data storage. Hence, the simulation outcome of the model was that it outperformed the scalability of other classical methodologies.

Li et al. [13] explored the model of Adaptive Network Coding (ANC) to increase the transmission performance. It obtained the best storage results compared with the traditional mechanisms. Additionally, the optimum value was obtained to appropriately determine the allocation for data storage. Due to this allocation, the data were distributed with x number of data components among X total data centers, which were also processed based on the probability factor. Consequently, the Optimal Storage Allocation (OSA) was utilized to rectify the existing problems. Both qualitative and quantitative analyses were performed, which proved the reliability of the data storage scheme. Assi et al. [14] utilized the Workload Assignment (WA) and Mixed-Integer Program (MIP) to derive a novel formulation for data storage. The problems in the WA were segregated into two sub-divisions: Latency-Aware Workload Assignment (LAWA-MIP) and Reliability-Aware Candidate Selection (RACS). Hence, the performance of the enhanced WA-Tabu model was measured using existing approaches and compared with the former methods. With the help of the simulation, they achieved the desired outcome to ensure the effectiveness of the cloud storage system.

Zaharie et al. [15] designed an automated data storage model in the cloud. It was designed based on the components used to allocate the machines to perform the services. This allocation was processed by considering the requirements of the software or hardware concepts. It aided in resolving the multi-objective issues while using the optimization algorithms and reduced the computational burden while storing the data. Techniques

such as Mathematical Programming (MP) and Optimization Modulo Theory (OMT) were considered for the experimentation. Hence, the existing challenges were mitigated by using the objective of search space reduction. The prime purpose of this model was (a) to achieve a symmetric value in a cloud system and (b) to represent in a graphical manner the functionalities related to specific constraints and (c) their association. In the last stage, the empirical results were measured with diverse metrics, and the efficiency was enhanced by obtaining the optimum values.

Santosh et al. [26] proposed a novel resource allocation algorithm for cloud computing based on the Reptile Search Algorithm (RSA), which mimics the hunting and encircling behaviors of crocodiles. The paper applied the algorithm to a healthcare application that involved Electrocardiogram (ECG) sensors and cloud data storage. The paper claimed that the algorithm can minimized the cumulative cost and improved the quality of service for cloud computing. Similarly, Haoyang et al. [27] presented an improved Ant Colony Optimization (ACO) algorithm for resource allocation in cloud computing for the new energy industry. The paper considered the characteristics and requirements of the new energy industry, such as load balancing, energy efficiency, and real-time response. The paper showed that the improved ACO algorithm could achieve better performance than the traditional ACO algorithm in terms of task completion time, number of iterations, and the load-balancing effect.

Suba and Sathya [16] developed the "Euclidean L3P-based multi-objective successive approximation (EMSA)" to ensure the security or privacy of the data. The data were non-resistant to such attacks; the role-aided encryption process was implemented. The novel EMSA was designed with the association of two traditional approaches, which were the Multi-objective Optimization Algorithm (MOA) and the Euclidean L3P Distance Algorithm (ELDA), as well as the Successive Approximation Iterative Proximate Algorithm (SAIPA). Finally, the efficacy of the recommended work was shown using three measures: privacy, fitness, and utility. From the analysis, the enhanced method achieved better privacy in storing the cloud data.

Shao et al. [17] considered the Provable Data Possession (PDP) model to ensure data integrity in the cloud sector. This model cannot ensure the privacy of the data since it is vulnerable to Third-Party Auditors (TPAs). It also depends on the process of the Public Key Infrastructure (PKI), for which the certificates become complex to manage. The "identity-based (IB) PDP" mechanism was introduced to overcome these limitations. Owing to the protocol, it struggled with the overhead problems, and the flexible and feasible aspects of the proposed system were used to reduce the complexities. Hence, the findings illustrated that the enhanced method achieved impressive results for the storage process. Akash et al. [18] gave a novel scheme to overcome the problem of allocating the storage nodes in a cloud environment. This was accomplished by representing a bipartite graph with the consideration of code blocks and holding thedata. Subsequently, the storage issue was diminished by using the concept of ant colony and greedy optimization, a clustering-aided approach. The experimental results proved that the implemented work obtained better results regarding displacement among the nodes, computational load, bandwidth, and disk space availability.

Ahmed et al. [28] presented a link-based penalized trust management scheme for preemptive measures to secure edge-based Internet of Things networks. They used a trust-based routing scheme that considered the load factor and the path history of the nodes to balance the traffic and avoid Denial-Of-Service (DoS) attacks. The benefit of this method is that it can improve network performance and reliability by selecting the most-trustworthy and least-vulnerable nodes for data communication. The limitation of this method is the potential for increased complexity in managing distributed systems and the need for efficient resource management, and it may require high coordination between the edge devices and the cloud server.

While Ijaz et al. in [29] discussed the need to enhance the traditional Advanced Encryption Standard (AES) algorithm to cope with emerging security threats in the cloud environment. The authors presented a framework with key features including enhanced security and the

owner's data privacy. The authors also introduced a trust and resource management scheme to improve the security and efficiency of the cloud services. The proposed framework modified the 128 AES algorithm to increase the speed of the encryption process, 1000 blocks per second, by the double-round key feature. The proposed algorithm involved less power consumption, better load balancing, and enhanced trust and resource management on the network. The simulation results showed that the proposed framework minimized the energy consumption by 14.43%, the network usage by 11.53%, and the delay by 15.67%.

Sangeetha and Sumathi [19] explored the modified random Fibonacci cryptographic (MRFC) technique using the encryption process with group key management, which resolved the storage problem in the cloud. Initially, the raw data were taken from standard sources. Further, the collected data were distinguished into non-sensitive and sensitive data groups. The sensitive and non-sensitive groups were mixed together for uploading in a private cloud for the encryption process. The prime contributions of the work were as follows: (i) despite considering machine learning, it utilized the sensitive attribute-based classification; (ii) the encryption was performed for sensitive data; (iii) the encryption time was reduced without affecting the data quality; (iv) the decryption took place with the required groups. Lastly, the simulation results were validated and analyzed with heterogeneous parameters such as the processing time, classification rate, security level, and storage space. Thus, the suggested work yielded better results in ensuring data privacy and data security when storing cloud data. It also achieved less cost for processing the functionalities.

Naseem et al. [30] presented an Artificial-General-Intelligence-based Rational-Behavior-Detection Agent (AGI-RBDA) for tracking online harm. They used various cognitive correlates such as intention, perception, motivation, emotions, and implicit and explicit knowledge to protect sensitive information like a human mind. They exposed and stimulated the behavior of different cognitive correlates and compared them with the encryption techniques used in expert systems. The benefit of this method is that it can analyze the intruder's behavior and overcome the encryption weaknesses in expert systems. The limitation of this method is that it may require high coordination between artificial intelligence and cognitive correlates.

Kashif et al. [31] presented a Network Functions Virtualization (NFV) for mobile core and heterogeneous cellular networks. They used a Software-Defined-Network-based Network Functions Virtualization (SDN-NFV) architecture to improve the performance and reliability of the Mobility Management Entity (MME) in 5G wireless communication systems. The benefit of this method is that it can reduce the cost and complexity of the MME deployment and enhance its scalability and flexibility. The limitation of this method is that it may require high coordination between the SDN controller and the MME endpoints. Similarly, Anwar et al. [32] presented a green communication framework for Wireless Body Area Networks (WBANs). They used an Energy-aware Link-Efficient routing approach (ELR-W) to select the best nodes for data transmission based on their energy level and the link quality.

Qaisar Shaheen et al. [33] discussed the importance of achieving energy efficiency in cloud computing. The authors presented a taxonomy of cloud computing energy/power management techniques and reviewed existing techniques based on a tool, OS, virtualization, and data center stages taxonomy. The benefits of energy-saving techniques include reduced energy consumption, cost savings, and improved sustainability. However, the limitations include the need for accurate modeling and monitoring of energy consumption, as well as the potential impact on performance and reliability. However, the same author in [34] provided a comprehensive analysis of the integration of Wireless Sensor Networks (WSNs) with cloud and fog computing. The benefits of the integration include the improved scalability, efficiency, and reliability of WSNs, as well as the availability of cloud and fog resources for data storage and processing. They also explored the integration of WSN/IoT with Fog Computing (FC) and found that WSN integration with FC has many benefits with respect to latency, energy consumption, data processing, and real-time data streaming. However, the WSN network still had some limitations in computing power,

storage resources, and battery life, which made the network restricted with respect to the data transformation.

Namdev et al. [35] proposed an optimized communication scheme for an energy-efficient and secure Flying Ad hoc Network (FANET) using drones and Unmanned Aerial Vehicles (UAVs). They used a Whale Optimization Algorithm based on Optimized Link State Routing (WOA-OLSR) over the FANET to provide optimal routing for energy-efficient and secure data transmission. Vijayalakshmi et al. [36] presented a novel intelligent channel estimation strategy for wireless communication systems using Multiple-Input Multiple-Output (MIMO) Orthogonal Frequency Division Multiplexing (OFDM). They combined the Chimp optimization and CatBoost algorithms to estimate and reduce the channel parameters. They used a Bi-directional Long Short-Term Memory (Bi-LSTM) network with an attention mechanism to jointly learn the sentiment analysis and sarcasm detection tasks.

These articles showed that optimization algorithms, machine learning techniques, and natural language processing can be applied to various domains of wireless communication systems to improve their performance.

## 3. Reliable Cloud Data Storage: System Model and Problem Formulation

### 3.1. System Model

Reliable cloud data storage has become the most-effective process for managing data. Cloud storage is the process of managing the data remotely and the process of safeguarding the data with third-party servers. To store the data in the cloud, the cloud can give the assurance to improve the security of the data. It considers the four distinct kinds of entities to achieve the storage mechanism. They are "the data owner, the data user, the cloud user, and the third-party server". The data owner manages the data to store them in various VMs. The data user can have the capacity to choose the machines where the data are recovered. Simultaneously, the third-party servers are used to check the data integrity frequently.

Storage mechanism: Cloud storage comprises many devices such as machines. Here, the data storage is nothing but mapping the logical and physical storage. Hence, considering the required components, the storage network may have several constraints while storing the data on the respective servers or VMs. Conversely, the storage process is differentiated into three types, which are explained as follows:

1. File storage: The files are hierarchically placed in this type. The information is stored in the metadata format of every file. Hence, the files are managed in higher-level abstraction types. Thus, it aids in improving performance.
2. Block storage: Here, the data or files are segmented into different chunks and represented with block addresses. This process does not contain the server for authorization.
3. Object storage: The encapsulation is performed with the object and metadata. Since the data belong to any type, they are distributed over the cloud. This also ensures the scalability and reliability of the system.

The major goals of designing a reliable cloud data storage system are listed below:

- Data reliability and availability: By storing the data with more machines or servers, the data user can obtain the encoded data to be deciphered further as the original data. When any of the servers has a fault, they are then used by the other effective servers, thereby enhancing the data integrity and reliability of the cloud network.
- Security: The better system enhances the security level. It also verifies the data integrity and confidentiality, which protects the network from any corrupted services.
- Offline data owner: Once the data are outsourced to a server or machine,there is no need to check the integrity of the stored data in the system.
- Efficiency: Due to this objective, the system's efficacy is reached in terms of less storage space, resolving the overhead problem in communication and computation, and so on.

Considering the above key points, the proposed reliable data cloud storage system using heuristic development is represented in Figure 1.

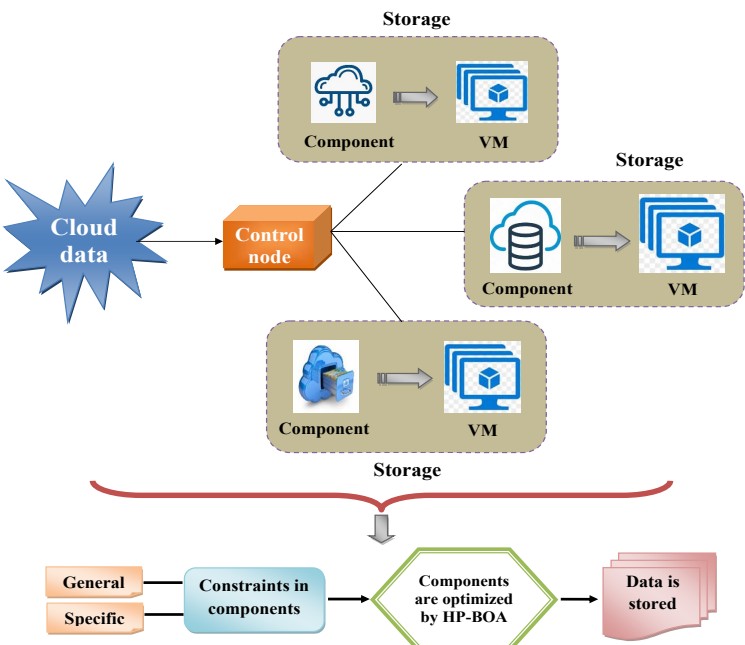

**Figure 1.** Architectural representation of proposed reliable cloud data storage using HP-BOA.

The primary aim of this novel framework is to save cloud data by rectifying the general and structural constraints. Firstly, it considers the components and VMs for storage purposes. Since the components pose various constraints, a new reliable model is introduced. Each piece of cloud data is specified with individual traits, which are then characterized by hardware and software components. Consequently, the new objective function is derived for solving both constraints. The interactions and data instances in the cloud data storage define the structural constraint. Similarly, the allocation rules and device capacity are included in the general constraints. To alleviate the constraint issues, a novel HP-BOA is newly proposed. In the last stage, the performance was measured with metrics, and its simulation results were carried out. Thus, the extensive results proved that the proposed work appropriately stored the cloud data using the components.

### 3.2. Problem Formulation

Cloud data storage is a system that refines storage and business accessibility. It arranges a vast number and various kinds of storage devices or machines, where the hardware or software components are considered. In the storage mechanism, the cloud is constituted by the data files $D$, the number of VMs $M$, and the number of components $C$. In addition to this, the storage mechanism is accomplished with security and integrity enhancement. When the cloud data are outsourced, some critical issues are faced in the mechanism, which degrade the quality of the services. Hence, the challenging issues and problems are described as given below:

1.  Deploying the cloud storage: While deploying the cloud storage, it relies on the requirements and technologies. Performing the process requires the geographical location. Hence, the storage cost becomes greater.
2.  Data virtualization: This is the process of mapping the data in the local storage to the physical storage, which includes the servers, storage, operating systems, and servers. This results in a higherruntime for storing the data.
3.  Data organization: This becomes cumbersome while arranging the data or dividing them into chunks, files, or blocks. Since it contains a poor format for storing, it loses its efficiency.
4.  Load balancing with data migration: When one of the machines faces a large load, some data are transferred to another machine. Yet, the challenges with the bandwidth or energy consumption exist.

5.  Machine constraints: Due to such constraints in machines or servers, critical data storage problems are faced. Factors such as the structural or general traits affect the storage process.

Therefore, the cloud network does not have the potential to control the stored data in its respective machines. Owing to this ineffective processes, the data can be destroyed, copied, altered, and so on. Moreover, the lack of control leads to imprecise results. While considering all the aforementioned problems, an effective, reliable cloud data storage was developed with the objective of heuristic development.

## 4. Specified Problem and Multi-Objective Function Derived for Reliable Cloud Storage

### 4.1. Problem Definition

Other than the above issues, cloud data storage mainly focuses on resolving the general and specific constraints in the components where the cloud data are stored effectively [15]. To define the issues in the reliable storage process, let us discuss the components and VMs in the cloud network. Here, the components are represented by $N = \{N_1, N_2, \ldots, N_C\}$ and VMs are denoted as $R = \{R_1, R_2, \ldots, R_M\}$. Every component is defined with requirement sets along with hardware traits. Similarly, each machine is composed of software or hardware features. These all come under the structural constraints among the components. Hence, the problems are listed below:

1.  An assignment matrix is constructed with the components and VMs, which are interpreted by each other. Hence, the entities in the matrix are either 0 or 1. This is given by Equation (1).

$$A_{xy} = \begin{cases} 1, & if \quad N_x \text{ is given to } R_y \\ 0, & if \quad N_x \text{ is not given to } R_y \end{cases} \tag{1}$$

2.  The type of every machine is denoted by its selection vector as $v_y$, where $y \in 1$ to $M$

Finally, all the constraints are: (i) the structural constraints; (ii) the hardware requirements (capacity constraints) of all components are satisfied; (iii) the purchasing/leasing price is minimized. The assigned matrix and its type selection are elaborated in Equations (2) and (3).

$$A = \begin{pmatrix} 0 & 0 & 0 & 0 & 1 & 0 \\ 0 & 0 & 1 & 1 & 0 & 0 \\ 0 & 1 & 0 & 0 & 0 & 0 \\ 1 & 0 & 0 & 0 & 0 & 0 \\ 0 & 1 & 1 & 1 & 0 & 0 \end{pmatrix} \tag{2}$$

$$v = \begin{bmatrix} 12, 12, 13, 13, 15, 0 \end{bmatrix} \tag{3}$$

Both the general and structural constraints are explored below.

General constraints:

While considering every component for allocating the data into any VMs, the allocation rules are specified concerning exclusive deployment. This is shown in Equation (4).

$$\sum_{y=1}^{M} A_{xy} \geq 1, \quad \text{where } x = 1 \text{ to C} \tag{4}$$

The capacity constraints consider solving the type of resource presented in the VM, which is derived using Equation (5).

$$\sum_{x=1}^{N} A_{xy} \cdot I_x^p \leq J_{y_v}^p, \quad \text{where } x = 1 \text{ to C} \tag{5}$$

One VM is linked with the data occupied by another VM. The term $\wedge$ is denoted as the wedge operator, which helps estimate two events' occurrence. Hence, the VM offers or links to store the cloud data are represented using Equation (6).

$$u_y = 1 \wedge v_y = 0 \Rightarrow Q_y^p = J_{v_y}^p \, \forall p \in (1, P) \wedge r_y = R_{v_y} \tag{6}$$

Furthermore, several VMs are used for deployment and estimation purposes. Here, $M$ represents the number of VMs used for deployment. Moreover, the variable $u$ determines the vector of the binary occupancy. It is defined by the vector value and given in Equation (7).

$$\sum_x^N A_{xy} = 0 \Rightarrow v_y = 0, \quad \text{where } y = 1 \text{ to M}$$

$$\sum_x^N A_{xy} \geq 1 \Rightarrow u_y = 1, \quad \text{where } y = 1 \text{ to M} \tag{7}$$

Structural or application-oriented constraints:

These two major constraints are taken as interactions among the components and instances, which are explained below:

- Conflict: When using the same VM for allocating the data, such conflicts are raised. The variable $\Im_{xy}$ is denoted as the conflicts among the two components encoded in the matrix $\Im$. The conflicts are in the matrix format shown in Equation (8).

$$A_{xy} + A_{zy} \leq 1, \quad \text{where } \Im_{xy} = 1 \tag{8}$$

- Co-location: This relation is given in the matrix when utilizing the same VM to define the constraints. Hence, $\aleph_{xy}$ represents the co-location relation stored in matrix $\aleph$. It is described in Equation (9).

$$A_{xy} = A_{zy}, \quad \text{where } \aleph_{xy} = 1 \tag{9}$$

- Exclusive deploying: From the different sets of components, only one is used for storage purposes. In that scenario, the constraint is defined by Equation (10).

$$\Re\left(\sum_{y=1}^M A_{x_1 y}\right) + \Re\left(\sum_{y=1}^M A_{x_2 y}\right) + \cdots + \Re\left(\sum_{y=1}^M A_{x_q y}\right) = 1 \tag{10}$$

Here, the function is referred to as $\Re(u)$, where the value becomes 1 if $u > 0$; otherwise, it is 0.

- Requirement is provided: The interaction constraint is considered when one component fetches some functionality from another. In turn, it creates instance constraints when interacting with the components. It is formulated using Equation (11).

$$p_{xy} \sum_{y=1}^M A_{xy} \leq q_{xy} \sum_{y=1}^M A \tag{11}$$

Here, $p_{xy}$ denotes the instance of components when consumed, whereas it provides the instances of one component declared by $q_{xy}$.

- Full deployment: This is achieved when the component is used to deploy in all leased VMs. It can be expressed using Equation (12).

$$\sum_{y=1}^M \left( A_{xy} + \Re\left( \sum_{z, \Im_{xy}=1} A_{zk} \right) \right) = \sum_{y=1}^M u_y \tag{12}$$

- Deploying with certain instances: This type of constraint occurs by generating the instances between components. Hence, the value should be greater, equal, or smaller. It is shown in Equation (13).

$$\sum_{x \in C} \sum_{y=1}^{M} A_{xy}\{op\}m \tag{13}$$

Here, *op* represents an equal, greater, or smaller value.

### 4.2. Objective Function Definition

The proposed model's Objective Function (*OF*) is discussed here. It is derived by two parameters: delay and penalty. To enhance the performance, the components were optimized with the aid of HP-BOA. The proposed work achieved a better outcome with the assistance of the optimized components. Therefore, the objective function is modeled using Equation (14).

$$OF = \arg_{\{N_C\}} \cdot \min\left\{ \sum_{y=1}^{M} u_y \cdot r_y \right\} \tag{14}$$

Here, $N_C$ is the number of optimized components to store the cloud data. Further, the term $u_y$ refers to the "binary occupancy vector". If the machine is used, its value is assigned as 1. Else, it becomes 0. Furthermore, $r_y$ signifies the VM offers or link types. Due to this objective function, it can mitigate the constraints presented in the components.

## 5. Hybrid Pelican–Billiards Optimization Algorithm for Reliable Cloud Storage System

### 5.1. Initialization and Solution Optimization

The solution is initialized and optimized using the proposed HP-BOA. The initialized data are taken as the solution, where the optimal solution is obtained to choose the best components. Here, the storage process is primarily performed by considering features and VMs, in which the component is represented as the data, hardware, or software. To perform the storage process, the proposed work took the different range of components as *C* and various counts of VMs as *M*. Among these presumed values, the components were optimized. The optimization was mainly performed based on the required components assigned to which VMs by using the proposed HP-BOA. Therefore, the solution is defined with the length of the total number of VMs. Due to this, it can resolve the constraints; thereby, it maximizes the performance rate. The diagram of the solution encoding is illustrated in Figure 2.

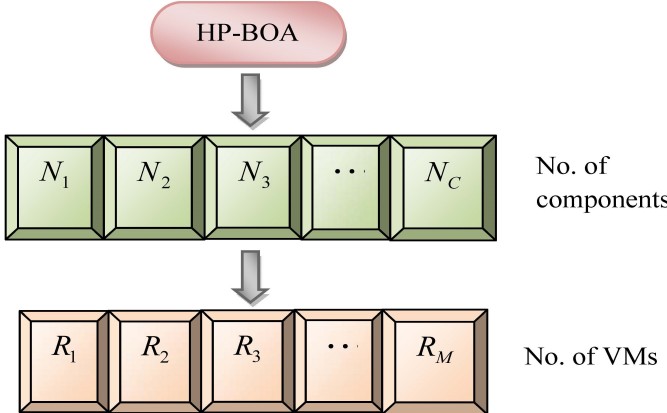

**Figure 2.** Solution encoding using HP-BOA.

### 5.2. Existing POA

The existing POA [37] was inspired by the natural behavior of pelicans, which perform the exploitation and exploration stages to obtain the optimum value. The mathematical expression of the POA is described below:

Stage 1: population initialization

The first step is to assume the required population in terms of pelicans and consider the total iteration count as $N$. This POA also contains the problem variables. When exploring the search space, the pelicans are arbitrarily initialized along with their corresponding lower- and upper-bound variables. It is expressed using Equation (15).

$$p_{a,b} = ld_b + rd \cdot (ud_b - ld_b), \quad \text{where } a = 1, 2, \cdots, \text{A}; b = 1, 2, \cdots, \text{B} \tag{15}$$

Here, the total count of pelicans and the problem variables are represented by A and B, respectively. Further, the upper- and lower-bound values of the $b^{th}$ problem variable are accordingly designated by $ud_b$ and $ld_b$. Then, the random value is varied from 0 to 1 in the form of $rd$. Finally, the $a^{th}$ pelican with the $b^{th}$ problem variable is defined by $p_{a,b}$. Further, the pelicans' objective function is calculated and given in Equation (16).

$$T = \begin{bmatrix} T_1 \\ \vdots \\ T_i \\ \vdots \\ T_A \end{bmatrix}_{A \times 1} = \begin{bmatrix} T(P_1) \\ \vdots \\ T(P_i) \\ \vdots \\ T(P_A) \end{bmatrix}_{A \times 1} \tag{16}$$

Stage 2: encircling the prey

After setting the valid parameters, all pelicans search for the prey location by moving forward. This process is performed through the random creation of the search space. With the help of the search space region, the pelicans also identify their prey's position, which is encircled. This encircling process can resolve the optimal locationissue. Hence, the pelicans move to reach the prey location, modeled using Equation (17).

$$p_{a,b}^{En} = \begin{cases} p_{a,b} + rd \cdot (tg_b - C \cdot p_{a,b}), & T_{tg}^{En} < T_a \\ p_{a,b} + rd \cdot (p_{a,b} - tg_b), & \text{otherwise} \end{cases} \tag{17}$$

In the aforementioned Equation (17), the new position is obtained, and it is denoted as $p_{a,b}^{En}$. The prey position is identified by $tg_b$. The exploration process is accomplished by deriving the objective function as $T_{tg}$ for the prey. Furthermore, $C$ signifies the random value of 1 or 2. When fixed as a value of 2, it acquires a greater distance between searching for and encircling the prey. Due to this objective derivation, it can potentially remove the non-optimal areas in the search space. Hence, it is formulated using Equation (18).

$$P_a = \begin{cases} P_a^{En}, & T_a^{En} < T_a \\ P_a, & \text{otherwise} \end{cases} \tag{18}$$

Here, the upgrading is performed with a new value for the pelican, which is denoted by $P_a^{En}$ with its corresponding objective function as $T_a^{En}$. Thus, the exploration phase provides the best value.

Stage 3: attacking the prey

After encircling the prey, pelicans attack or hunt the prey. This is performed by the pelicans spreading their wings alongthe water's surface. The process leads to the prey (fish) moving upwards, where they are attacked. Hence, this exploitation process achieves a

high convergence speed to produce good results. Equation (19) represents the derivation of attacking the prey.

$$p_{a,b}^{Ex} = p_{a,b} + S \cdot \left(1 - \frac{n}{N}\right) \cdot (2 \cdot rd - 1) \cdot p_{a,b} \tag{19}$$

Here, the new position is obtained in the exploitation phase, which is indicated by $p_{a,b}^{Ex}$, concerning the $b^{th}$ dimension of the $a^{th}$ pelican. The term refers to the constant value fixed at 0.2; then, the POA's current and total iteration are denoted as $n$ and $N$, respectively. The coefficient $S \cdot \left(1 - \frac{n}{N}\right)$ is used to represent the radius where the global solution is acquired. In addition, the efficient expression determines whether the pelican's new solution is accepted or ignored. This is given in Equation (20).

$$P_a = \begin{cases} P_a^{Ex}, & T_a^{Ex} < T_a \\ P_a, & \text{otherwise} \end{cases} \tag{20}$$

Finally, the optimal position and new objective function are denoted by using $P_a^{Ex}$ and $T_a^{Ex}$, respectively. Hence, with the implementation of the exploration and exploitation phase, the optimum value is obtained, which helps optimize the machines and components for storing the data over the cloud network. The pseudo-code of the conventional POA is given in Algorithm 1.

---

**Algorithm 1** Traditional POA.

---

1:  Assume $P$ as pelican population and total iterations as $N$
2:  Let $B$ be the problem variable
3:  Compute the fitness value
4:  Derive the objective function
5:  **while** $(n < N)$ **do**
6:      Evaluate the fitness value
7:      Determine the location of the prey
8:      **for** each billiard ball **do**
9:          **Exploration phase:**
10:         **for** every problem variable **do**
11:             Find the new location using Equation (17)
12:         **end for**
13:         Using Equation (18), the new status is generated
14:         **Exploitation phase:**
15:         **for** every problem variable **do**
16:             Using Equation (19), the new position is upgraded
17:         **end for**
18:         Using Equation (20), the new status is determined
19:      **end for**
20:      Update the best solution
21:      $n = n + 1$;
22: **end while**
23: Obtain the best value

---

### 5.3. Existing BOA

The traditional BOA [38] was inspired by the mechanism of billiard games. It is mainly used to enhance performance by providing parameter optimization. It is performed by considering several pockets, billiard balls, and threshold and decision variables. The following steps explore the mathematical expression of the BOA.

Stage 1: initialization

Firstly, the required balls are initialized in the search space, which is arbitrarily distributed to achieve the optimum value. Hence, it is derived using Equation (21).

$$D_{a,b}{}^0 = X_b^{mn} + rd \cdot (X_b^{mx} - X_b^{mn}) \tag{21}$$

In the above equation, $a = 1, 2, 3, \cdots, A$ and $b = 1, 2, 3, \cdots, B$. Here, the total population of balls is denoted as A, and the total variables are defined by $B$. The first value of the $a^{th}$ billiard ball, along with the $b^{th}$ variable, is declared as $D_{a,b}^0$. Further, the minimum and maximum variable values are denoted as $X_b^{mn}$ and $X_b^{mx}$ irrespective of the $b^{th}$ variable, correspondingly. The random value $rd$ lies in the range of $[0, 1]$. The position of billiards and balls is determined using an objective function. Over the iterations, the location is upgraded with the best solution. After the pockets are considered, the balls are distinguished into cue and ordinary balls.

Stage 2: revising the ball position

Owing to more balls, collisions occur in this algorithm. After a collision, a new position is obtained to upgrade the balls to reach the pockets. Hence, the position relies on the shot's precise value. By the objective of the exploitation process, the error is decreased in the search space. Then, the new position is derived using Equation (22).

$$D_{a,b}^{new} = rd \cdot (1 - NR) \left( D_{a,b}^{old} - Y_{c,b}^a \right) + Y_{c,b}^a \tag{22}$$

The new position and old position of the $b^{th}$ variable of the $a^{th}$ ordinary ball are indicated by $D_{a,b}^{new}$ and $D_{a,b}^{old}$, respectively. The solution acquired that comes under the $b^{th}$ variable of the $c^{th}$ pocket along with the $a^{th}$ pair of ordinary balls is signified by $Y_{c,b}^a$, where $a = 1, 2, 3, \cdots, A$. Simultaneously, the random value as $rd$ is defined as the error rate between $[-Er, Er]$. Finally, the term $NR$ represents the precision rate calculated by Equation (23).

$$NR = \frac{T}{T_{\max}} \tag{23}$$

The current and maximum iteration values are denoted by $T$ and $T_{max}$.

Stage 3: velocity computation

Once all the balls have collided, the velocity value updates the ball's position. Further, the velocity is derived using Equation (24).

$$L_a{}' = \sqrt{2c \cdot \overrightarrow{D_a^{old} D_a^{new}}} D_a^{old} D_a^{new} \tag{24}$$

After the collision, the ball's velocity is designated as $L_a{}'$. The velocity of the $d^{th}$ ordinary ball is denoted as $V_d{}'$. Further, the movement vector is created through $\overrightarrow{D_a^{old} D_a^{new}}$, and the unit movement vector is noted by $D_a^{old} D_a^{new}$. Consequently, the acceleration rate is signified by $c$, which is 1. Hence, the velocity is estimated for cue balls before and after the collision, which is given in Equations (25) and (26).

$$L_{a+A} = \frac{L_a{}'}{D_a^{old} D_a^{new} \cdot D_{a+A}^{old} + A \cdot D_{a+A}^{old}} D_{a+A}^{old} D_d^{old} \tag{25}$$

$$L'_{a+A} = \chi \cdot \left( 1 - \frac{T}{T_{\max}} \right) \left( L_{a+A} - L_a{}' \right) \tag{26}$$

In the above equations, the velocity evaluated before the collision is denoted by $L_{a+A}$, as well as for after the collision is indicated by $L'_{a+A}$. Further, the term $\chi$ refers to the user-defined variable in the $[0, 1]$ range. With the attainment of the velocities, the updating is performed through Equation (27).

$$D_{a+A}^{new} = \frac{L'_{a+A}}{2c} L'_{a+A} + L_a^{old} \tag{27}$$

Stage 4: threshold for local optima

To find the best value, the threshold value $H$ is considered to find the best value, which is used to alter the variables. This threshold is compared against a random value as $rd$, which varies from 0 and 1. When it meets $rd < H$, the position updating is performed by Equation (28).

$$D_{a,b}^0 = X_b^{mn} + rd \cdot (X_b^{mx} - X_b^{mn}) \tag{28}$$

This process is repeated until the stopping criterion. The pseudo-code of the classical BOA is elucidated in Algorithm 2.

---

**Algorithm 2** Classical BOA pseudo-code.

---

1: Set the initial population in terms of billiard balls
2: Consider the parameters
3: **while** ($n < N$) **do**
4:     The ball and pocket positions are determined and modeled with an objective function
5:     Generate the ordinary and cue balls
6:     Using Equation (22), the location of the ordinary ball is upgraded
7:     The velocity is computed using Equation (24)
8:     Before and after the collision, the velocity updating is performed by Equations (25) and (26)
9:     The new position is generated using Equation (27)
10:     **if** $rd < H$ **then**
11:         The size of the ball is altered with Equation (28)
12:     **end if**
13:     $n = n + 1$;
14: **end while**
15: Acquire the best value

---

*5.4. Proposed HP-BOA*

The new algorithm is proposed combining the POA and BOA, respectively. Though these two algorithms contain vital points, they have some significant limitations. During the exploration phase of the POA, the obtained position does not satisfy the proposed concept. As it fails due to the optimal global solution issues, it does not deliver the best solution to rectify the constraints in the components for the proposed storage model. Therefore, the position value of the exploration phase is considered for further upgrading. In the BOA, the collisions tend to degrade the performance when it is handling a massive amount of data or real-time data. Furthermore, every ball varies with velocity. Thus, it cannot deliver the best value. Hence, the position based on the velocity is taken. To surmount these issues, a new position update was accomplished by developing the HP-BOA. Firstly, the two positions, i.e., Equations (19) and (27), determine their difference. This is formulated using Equation (29).

$$Diff = p_{a,b}^{Ex} - D_{a+A}^{new} \tag{29}$$

The term $p_{a,b}^{Ex}$ signifies the position in the POA, and the position of the BOA is denoted by $D_{a+A}^{new}$. With the help of the difference value, the final position is calculated and given in Equation (30).

$$Po^{final} = \frac{Po^{final} + Diff}{2} \tag{30}$$

Thus, the optimal value is obtained by the above equation. The pseudo-code of the proposed novel HP-BOA is explained in Algorithm 3, and its flowchart diagram is given in Figure 3.

---

**Algorithm 3** Proposed HP-BOA pseudo-code.

---

 1: Assume the population as pelicans, as well as billiard balls
 2: Set the parameters
 3: Determine the objective function
 4: **while** ($n < N$) **do**
 5:     Position is updated using Equation (19) in the POA
 6:     Position is updated using Equation (27) in the BOA
 7:     Estimate the difference between the two positions using Equation (29)
 8:     Final position is revised using Equation (30)
 9:     $n = n + 1$;
10: **end while**
11: Obtain the best value

---

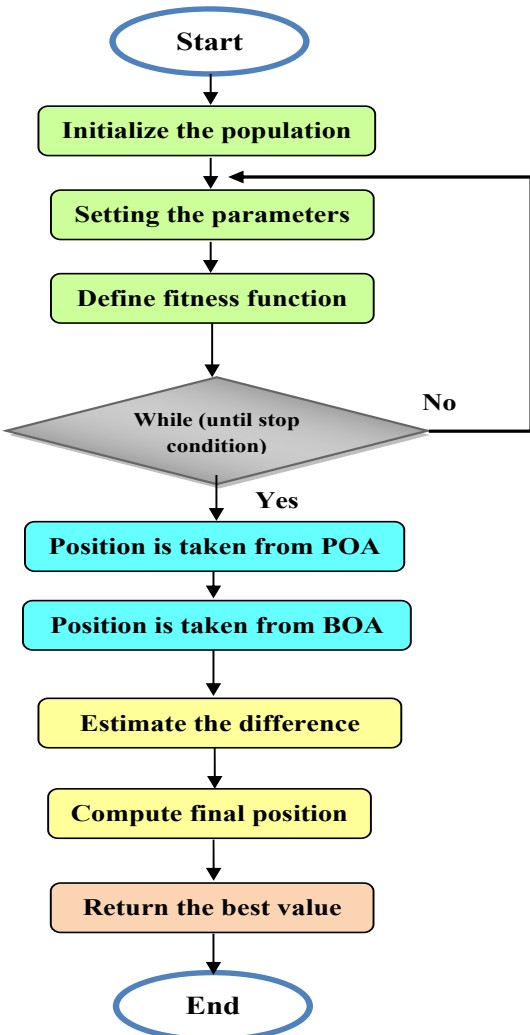

**Figure 3.** Flowchart diagram of proposed HP-BOA.

## 6. Result Analysis

### 6.1. Experimental Setup

The proposed reliable data storage was simulated using MATLAB 2020a, and the simulation analysis was carried out. We provide the convergence, comparative, and statistical analysis to validate the model's effectiveness. The terms active servers, computational time, and makespan were taken. We also considered 10 for the population number and the maximum iterations as 100. The enhanced system was compared with HHO [39], the FFA [40], the POA [37], and the BOA [41].

### 6.2. Performance Metrics

Three main measures were used, and their description is given below:

1. Active server: As the name implies, the number of servers was considered for the storage over the cloud environment.
2. Computation time: This is the time elapsed required for storing the cloud data in the respective machines based on the components.
3. Makespan: This is "the length of time that elapses from the start of work to the end". The time was calculated at the beginning stage of storing the data until they was fed into the components.

### 6.3. Configuration Setting

The proposed reliable system constitutes four configurations encompassing the components and VMs. Hence, Table 2 illustrates the configuration setting for storing the cloud data.

**Table 2.** Configuration setting of proposed cloud data storage.

| Configuration Case | No. of Components | No. of VMs |
| --- | --- | --- |
| 1 | 6 | 5 |
| 2 | 12 | 10 |
| 3 | 18 | 15 |
| 4 | 24 | 20 |

### 6.4. Convergence Analysis of Cloud Data Storage System

The convergence analysis of the recommended model is presented in Figures 4 and 5, when compared against heuristic algorithms concerning the four configuration cases. Similarly, Figure 5 depicts the convergence results of the model that contains 18 components and 15 VMs for storing the data. The obtained values of 0.018% of the HHO, 0.0055% of the FFA, 0.01% of the POA, and 0.003% of the BOA were more than the newly suggested HP-BOA at the 20th iteration. Hence, it can maximize the convergence rate. Based on the outcome, it ensures the data are stored in the machines properly.

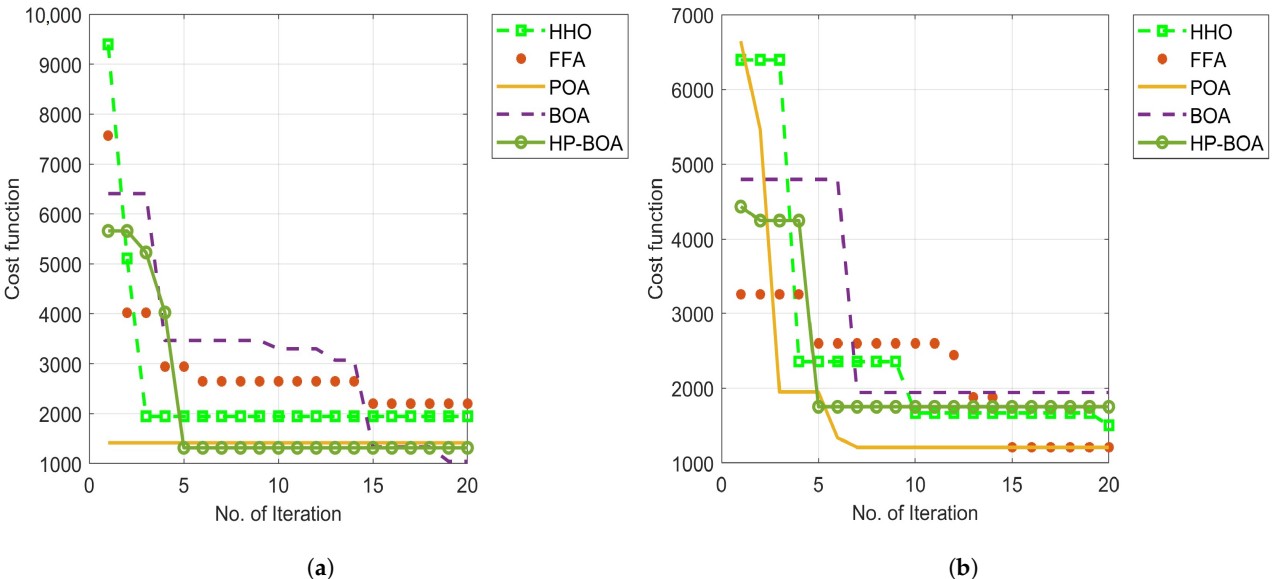

(a)                    (b)

**Figure 4.** Convergence analysis of the proposed reliable cloud data storage using the HP-BOA regarding various cases: Configuration Case 1 (**a**) and Configuration Case 2 (**b**).

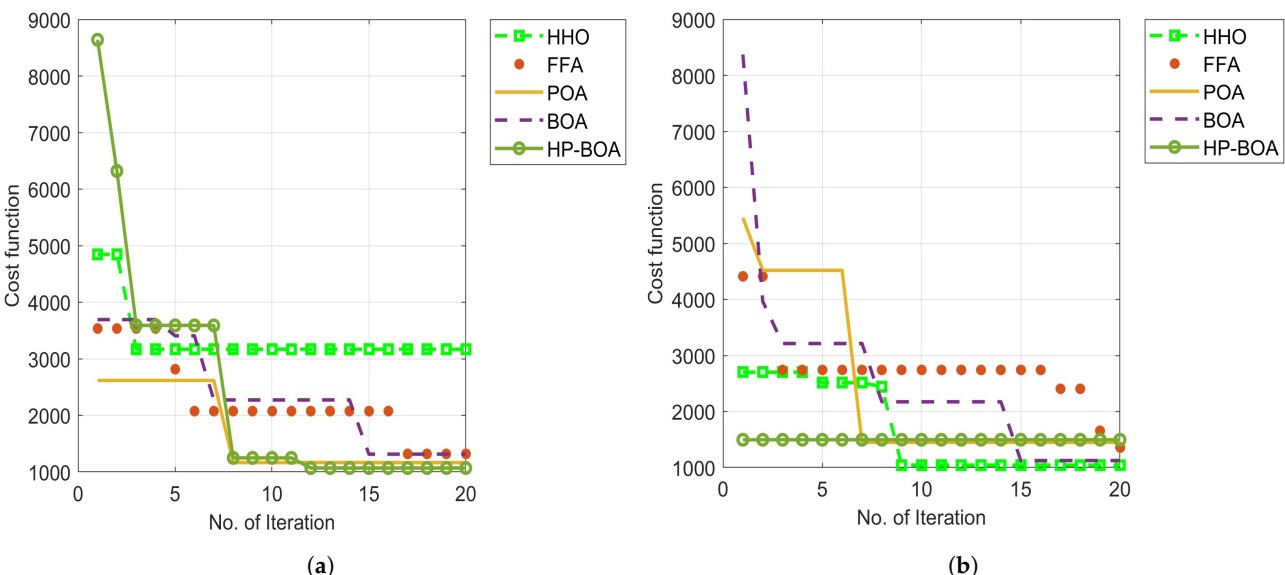

**Figure 5.** Convergence analysis of the proposed reliable cloud data storage using the HP-BOA regarding various cases: Configuration Case 3 (**a**) and Configuration Case 4 (**b**).

*6.5. Active Server Analysis of Cloud Data Storage System*

Figure 6 shows the active server comparative analysis of our reliable model. Compared to the other approaches, the proposed model required fewer active servers. Due to this, it can save the energy required for storing cloud data. Thus, the reliability performance was improved.

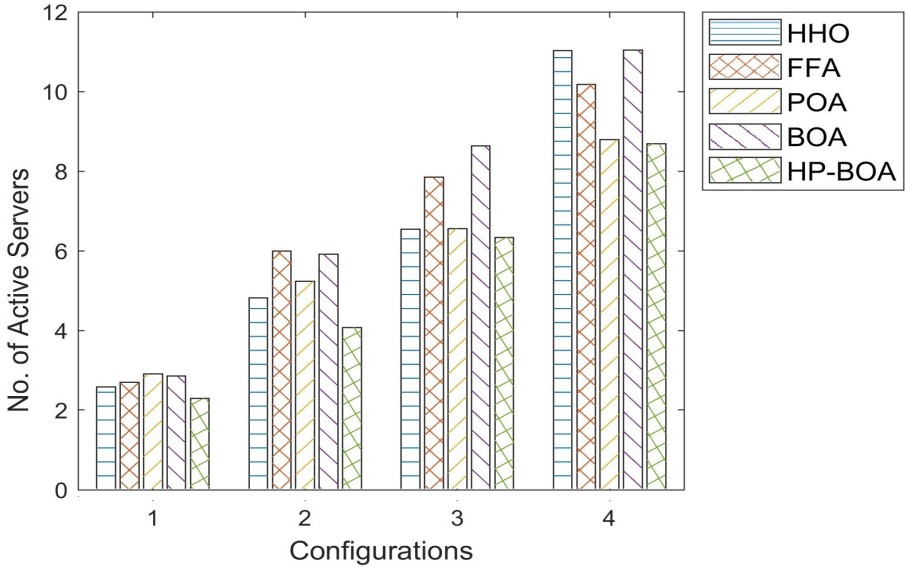

**Figure 6.** Analysis of the number of active servers for the proposed cloud data storage model with various configuration cases over the classical algorithms.

*6.6. Computation Time Analysis of Cloud Data Storage System*

Figure 7 illustrates the computational time analysis compared to the other heuristic algorithms. In the second configuration, the existing FFA had a high computational time. Hence, processing in the cloud data storage model took considerable time. Moreover, HHO had the second-best performance. At each configuration, the designed model showed a lower computational time. Hence, it provided better performance in the cloud

data storage model. Therefore, it proved that it could reduce the time complexity and enhance the performance.

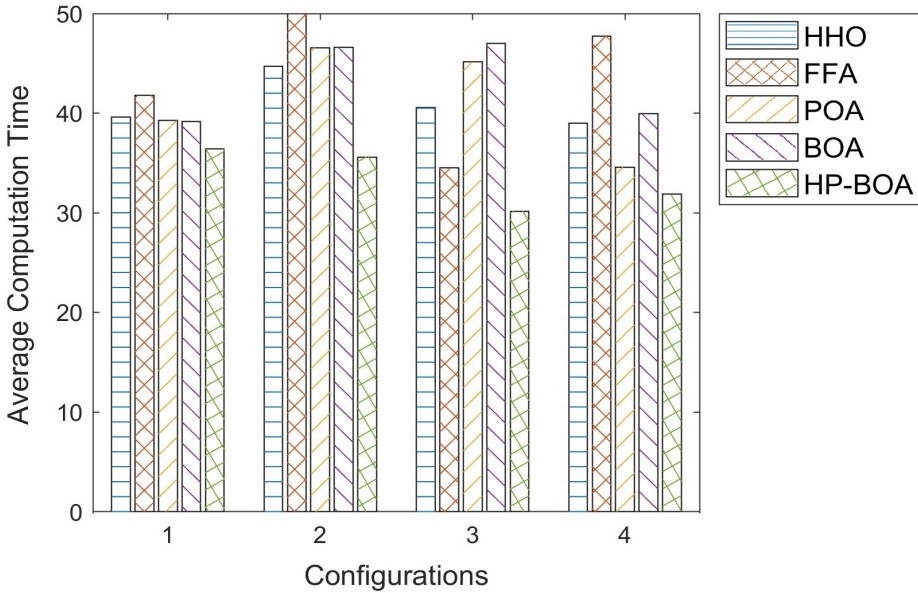

**Figure 7.** Analysis of the computational time of the proposed cloud data storage model with various configuration cases and classical algorithms.

### 6.7. Makespan Analysis of Cloud Data Storage System

The makespan analysis of the recommended work is given in Figure 8 provides the makespan with analysis of the recommended work. In the fourth configuration scenario, the makes 84% less than the FFA and POA, and 21.15% less than the BOA, respectively. Hence, the model's outcome demonstrated that it has less required time for processing to store the cloud data efficiently.

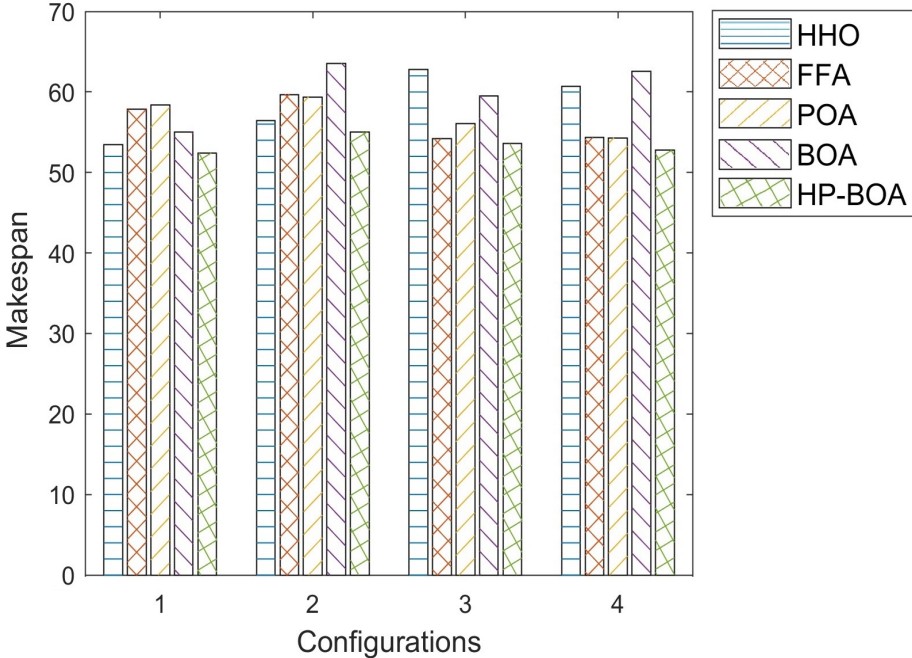

**Figure 8.** Analysis of the makespan of the proposed cloud data storage model with various configuration cases and classical algorithms.

*6.8. Statistical Evaluation of Cloud Data Storage System*

The statistical analysis of the proposed model is given in Table 3. This estimation was processed using factors such as "best, worst, median, mean, and standard deviation". The mean is the average value of the best and worst values, and the median is the center point of the best and worst values. The standard deviation is the degree of deviation between each execution. The proposed model achieved impressive results from the findings to satisfy the requirements.

**Table 3.** Statistical evaluation, based on the percentage (%) value, of the reliable cloud data storage compared over the conventional algorithms.

| Metrics | HHO [39] | FFA [40] | POA [37] | BOA [41] | HP-BOA |
|---|---|---|---|---|---|
| Configuration Case 1 | | | | | |
| Best | 394 | 396 | 391 | 393 | 390 |
| Worst | 9394 | 7565 | 1416 | 6407 | 5660 |
| Mean | 1318 | 1470.1 | 1211.9 | 1288.9 | 1100.7 |
| Median | 1024 | 1005 | 1416 | 1044 | 1258 |
| Standard deviation | 994.31 | 1006.9 | 410.32 | 1235.5 | 940.81 |
| Configuration Case 2 | | | | | |
| Best | 399 | 391 | 391 | 398 | 380 |
| Worst | 6394 | 3256 | 6643 | 4797 | 4432 |
| Mean | 1399.8 | 1326 | 1138.6 | 1447.6 | 1283.2 |
| Median | 1205 | 1211 | 1063 | 1317 | 1347 |
| Standard deviation | 983.39 | 624.81 | 780.01 | 1014.3 | 801.4 |
| Configuration Case 3 | | | | | |
| Best | 380 | 341 | 360 | 380 | 330 |
| Worst | 4846 | 3535 | 2618 | 3693 | 8641 |
| Mean | 1554.4 | 1397.7 | 1091.4 | 1229.9 | 1074.6 |
| Median | 1032 | 1319 | 1120 | 1140 | 1069 |
| Standard deviation | 1031.8 | 606.98 | 514.42 | 818.39 | 1163.9 |
| Configuration Case 4 | | | | | |
| Best | 396 | 395 | 399 | 398 | 390 |
| Worst | 2701 | 4409 | 5456 | 8372 | 1495 |
| Mean | 1146.1 | 1398.5 | 1213.6 | 1129.1 | 947.01 |
| Median | 1040 | 1131 | 1252 | 1111 | 1120 |
| Standard deviation | 445.34 | 759.31 | 976.67 | 1053.2 | 500.62 |

## 7. Conclusions

This research presented a novel reliable cloud data storage system using a hybrid heuristic algorithm. As the existing algorithms consider the components and VMs for storing the data, they fail with respect to the general and specific constraints. The general constraints were defined as the instances and interactions among the components. Further, the constraints were identified with allocation rules and the device's capacity. To combat such constraints, the novel HP-BOA was proposed. With the help of the HP-BOA, the components were optimized. Thus, it could resolve the issues while storing the cloud data in the components. Finally, the performance was validated and compared with the classical models. A comparative and statistical evaluation was performed to ensure the system's effectiveness. While using the third configuration case, the computational time was 36.6% for HHO, 13.3% for the FFA, 50% for the POA, and 56.6% for the BOA, respectively, which were more than that of the HP-BOA. Thus, the proposed system yielded an empirical value that provided adequate storage performance.

**Author Contributions:** All authors contributed equally to the research and wrote the article. The conceptualization, methodology, and writing—original draft were performed by A.T. The supervision and direction are performed by F.C. and Z.M. Q.S., A.A. and B.H. performed the detailed investigation, formal analysis, and coordination. The funding acquisition and resources were provided by K.-I.K. B.H.L. took part in the writing—reviewing, and editing. All authors have read and agreed to the published version of the manuscript.

**Funding:** This research received no external funding.

**Institutional Review Board Statement:** Not applicable.

**Informed Consent Statement:** Not applicable.

**Data Availability Statement:** No new data were created in this study. Data sharing is not applicable to this article.

**Acknowledgments:** This work was supported by Chungnam National University.

**Conflicts of Interest:** The authors declare no conflict of interest.

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
