# Peer review of "Hybrid HP-BOA: An Optimized Framework for Reliable Storage of Cloud Data Using Hybrid Meta-Heuristic Algorithm"

_applsci, doi:10.3390/app13095346_

Round 1
Reviewer 1 Report
This paper is well-organized except some minor concerns.
1) The texts in Fig.1 is too small to read.
2) The texts in all figures shall be in the same font.
3) Please add patterns on the histograms in Fig.6~8 (no just colors) to differentiate them.
4) What is the unit of the numbers in Table 3?
Author Response
Dear Reviewer,
Thank you for your valuable comments concerning our manuscript entitled “Hybrid HP-BOA: An Optimized Framework for Reliable Storage of Cloud Data using Hybrid Meta-Heuristic Algorithm” for further improvements. We have carefully read the manuscript and made corrections based on the comments mentioned here. Your recommendations were found to be very useful for the current version as well as for future considerations. The corrections marked in red color are now revised in the manuscript.

Reviewer 2 Report
Dear authors,
I have finished the review of your paper. In my opinion you can include at least one example of your results tested using a pricing calculator offered by one o more public cloud providers (Microsoft, Amazon or Google).
You can also provide an example of a cloud application based on your proposed results.
Hope you find these recommendations useful.
Author Response

(The authors gave the same response as above.)

Reviewer 3 Report
The authors propose an optimization method for storing cloud data using the hybrid pelican-billiards optimization algorithm. Issues and problems with features and reliable data storage are presented as background information to the consideration of the HP-BOA for reliable cloud storage system.
On pages 16-18, the figures depict convergence, number of active servers, computation time, and makespan.
Questions:
1. As most of the differences occurs less than 20 iterations in Figure 4, is it possible to adjust the figures to expand the depiction of what is happening between 0 and 20 iterations by limiting the number of iterations shown in the graphic? This would allow the reader to make a comparison more easily. Could the lines be depicted with shapes such as one line be a solid line and the others be combinations of dashes, spaces, and dots (e.g., _ _ _, _ . _ . _, _ .. _ .._, etc.)Those viewing the figures in gray scale would not be able to differentiate the line source.
2. The cost function calues in Figure 5 appear to be less for HHO and FFA compared to HP-BOA at the 20th iteration. How are the percents calculated in section 6.4? Is “The obtained value” the cost function value per iteration or the cost function value at 20 iterations? This needs to be explained for the reader to understand the comparisons.
3. For figures 6, 7, and 8, although the colors are differentiated if in color, gray scale printing and viewing will be a challenge. Is it possible to shade using the same color from light to dark so that they are graduated with HHO being the lightest in color and HP-BOA as the darkest?
4. Section 6.5, the HP-BOA approach requires less active serves for each of the configurations. This is clearly stated. Are less active servers and less energy a reliability element? Energy as presented in the paper earlier is an element of optimization for functionality rather than reliability. Consider updating this sentence to describe why lower energy costs increases reliability as well as optimizes resource utilization.
5. Section 6.6, how are the percentages determined? The HP-BOA appears to be around 35 time units whereas HHO is 45, FFA is 50, POA and BOA around 47. It seems that relative to the amount of the HP-BOA time, that HHO requires 28.5% more time (28.5%of 35 = ~10), FFA requires 42.8% more time (42.8% of 35 = ~15), and POA and BOA require 34.2% more time (34.2% of 35 - ~12). The percentages are correct; however, how they are stated is confusing. Suggest editing how the percentages are presented to indicate they are of the time for HP-BOA during the second configuration. (Look at how section 6.7 is presented with the lesser than statements).
6. Table 3 – best and worst values is stated. Are the values a reliability measure? Which equation is used to calculate these best, worst, mean, median, and standard deviation values? By knowing what equation or what the values is measuring would help with the comparison and understanding of why the configuration with the lowest standard deviation is not the best solution when the best, worst, mean, and median values are the same.
7. Similarly to section 6.6, in the conclusions, the relativity of the percentages presented require qualification as less than or greater than. As presented, 36.6% of HHO would be 36.6% of 45 = 16.5. The difference between HHO and HP-BOA is about 10 units of computational time.
Thank you for this interesting presentation or cloud storage optimization challenges.
Author Response

(The authors gave the same response as above.)

Reviewer 4 Report
(1) It is not recommended to list the year in the "Literature Review", see lines 86-152, and all "etal" in the article should be "et al.
(2) In Section 3.1, the authors should provide a detailed description of the system model.
(3) Many capitalized terms in Section 2.2 should have their corresponding abbreviations, e.g. Adaptive Network Coding, Optimization Modulo Theory, etc. In addition, some abbreviations are not written in capital letters, such as PSO, EMSA, etc. Please review the contents of lines 154 - 172 in detail.
(4) Equations 6 and 7 need to be fully described, what mathematical operations are denoted by ∧ and ⇒ respectively? It is suggested that authors use the correct mathematical expressions for the expressions.
(5) What do Sxy and Nxy mean in Eq. 8 and Eq. 9, respectively? The authors have not explained? This formula needs serious revision.
(6) What is OP in eq. 13? The authors' formula has many problems and needs to be revised.
(7) Could the authors please check if equation 24 is correct?
(8) Is the expression [0, 1] in line 405 correct?
(9) In line 456, the authors compare complexity? How is this complexity measured? Is there a computation metric, e.g. computed FLOPs?
(10) Why do the authors compare HHO and FFA in the experimental section, needs to be clarified.
(11) POA and BOA should be written similarly to HHO and FFA in detailed pseudocode and working procedure. Please add these two algorithms before Section 5.4.
(12) The data in Table 3 are very confusing? Why is the WORST of all 5 algorithms the same under the four cases? This needs to be explained. There are also other metrics that are the same in many cases, and the authors should clearly explain them.
In summary, this article is very rough and does not meet the publication standards. Please submit it for review after the author addresses the above concerns.
Author Response

(The authors gave the same response as above.)

Round 2
Reviewer 4 Report
I have no further questions about this paper. I think it should be considered for publication.
Author Response
Dear Reviewer,
We are very much thankful for your valuable consideration and recommendations of our manuscript entitled “Hybrid HP-BOA: An Optimized Framework for Reliable Storage of Cloud Data using Hybrid Meta-Heuristic Algorithm” for publication.